# The Mobile Emotional Intelligence Test (MEIT): An Ability Test to Assess Emotional Intelligence at Work

**Martin Sanchez-Gomez \* and Edgar Breso \*** 

Department of Evolutionary, Educational, Social Psychology and Methodology, Universitat Jaume I, Castellón, 12071 Castellón de la Plana, Spain

\* Correspondence: sanchgom@uji.es (M.S.-G.); breso@uji.es (E.B.)

**Abstract:** The present study analyzes the Mobile Emotional Intelligence Test (MEIT), a new ability-test to assess emotional intelligence (EI) in a digital way. Taking into account the importance of emotional competencies in the study of employees' wellbeing and performance, the instrument tested is based on the most supported ability model (Four-branch Mayer and Salovey Model), and it evaluates emotional capacity through nine different emotional tasks. A total of 1549 participants (841 women and 708 men) with an average age of 27.77 (SD = 8.75) fulfilled the MEIT, consisting of 42 items. The score on the test is based on expert judgments: professional psychologists and emotional intelligence specialists. In addition to the MEIT test, a series of questionnaires was used to assess relevant constructs which research has shown to be related to EI (general intelligence, personality traits, and life satisfaction); besides, another measure of emotional intelligence trait (TMMS-24) was included. The results showed that the MEIT is a reliable and valid test that is useful for both scientific research and individual assessment. Statistical analysis provides evidence of the reliability and validity of the three-factor structure of the questionnaire. Moreover, internal consistency measures were high. In line with previous studies, MEIT maintains the expected relationships with the rest of the constructs studied. Finally, the limitations of the present study and the need for future research on emotional intelligence assessment are discussed.

**Keywords:** MEIT; emotional intelligence; ability test; mobile; digital; validation

## 1. Introduction

### 1.1. Workplace Health

Work is fundamental to economic and psychological wellbeing for individuals and for society overall. Health promotion in workplaces has become a central feature of health policy in many countries due to the epidemic in chronic diseases and the ageing population. Different findings offer a baseline for new initiatives related to promote healthy workplaces and healthy individuals by improving conditions [1–4]. Moreover, it has been demonstrated that workplace health promotion programs have a positive return on investment (ROI) [5]. Therefore, it is necessary to attend to psychological aspects to improve and dignify workplaces. For that reason, the present study implies an advance in the assessment of Emotional Intelligence (EI), considering that it is a key competence in interpersonal relationships in work places and a basic ability for future enterprises.

### 1.2. Emotional Intelligence Concept

Organizations increasingly seek to maximize labor performance, and thus they focus on so-called 'soft' abilities, among which EI is prominent.

The concept of EI was established 25 years ago, and since then it has been developed, gaining popularity and visibility among researchers and professionals. According to the definition suggested by Salovey and Mayer [6], EI is "the subset of social intelligence that implies the ability to monitor one's own and others' feelings and emotions, to discriminate among them and to use this information to guide one's thinking and action" (p. 189).

The mentioned model is currently the most supported scientifically in the study and practice of EI worldwide. It is an integrative model that understands EI as a combination of four different skills: (1) Adequate perception of one's own and others' emotions; (2) Emotional facilitation of thinking; (3) Understanding of one's own and others' emotions; and (4) Proper management of emotions to achieve a specific aim. In this way, EI is considered a set of skills ordered hierarchically from an inferior level (emotion perception) to a superior level of complexity (management). These intertwined skills influence people's ability to interact with others in a proper manner [7], to communicate in an effective way [8], to manage conflicts [9], to manage stressful situations [10], and to create a positive labor environment [11], among many other aspects. After years of research, it has been assumed that these skills can explain important outcomes such as social relationships, illegal drug and alcohol use, and deviant behavior or salary [12–14], and it is therefore necessary to come up with adequate measurement instruments for these abilities.

### 1.3. Emotional Intelligence and Its Measure

So far, two different ways of assessing EI have been developed [15]: The first one is based on the use of self-report, through which it is expected that people expose how well they perceive and manage emotions. According to this, EI is measured by means of self-perceptions, using items that ask questions such as: "I usually spend time thinking about my emotions" or "I pay much attention to feelings". One of the most widely used questionnaires in research has been the Trait Meta-Mood Scale (TMMS) developed by Salovey, Mayer, Goldman, Turvey, and Palfai [16], which assesses the levels of interpersonal emotional intelligence through three factors: attention to feelings, emotional clarity, and mood repair. These questionnaires intend to inform in a precise way about the actual abilities; hence, a very precise introspection is required, which is unusual for most people [17]. In general, people wrongly estimate their own levels of intelligence, either their general intelligence or EI [13]. Besides, most of the EI measures through self-reports are influenced by the interviewee's self-esteem and mood [18].

The second form of evaluating EI consists on using skill tests based on performance criteria. These tests are more objective than self-reporting, however they demand more time and effort, both process structuring and administrative-wise [19]. Another inherent difficulty in skills tests corresponds to the score criteria. Facing the difficulty of assigning emotional response suitability, the score procedure is based on agreement, which means that the most frequent answer is considered the most correct one, or according to experts [20]. The most widely used ability measure has been the Mayer–Salovey–Caruso Emotional Intelligence Test (MSCEIT) developed by Mayer, Salovey, and Caruso in 2002 [21].

### 1.4. Mobile Emotional Intelligence Test (MEIT): A New Way of Evaluating Emotional Intelligence (EI)

Although numerous measurement instruments have already been created [22–26], each of the existing instruments has its disadvantages. The deficiencies of these measures are diverse; there is remarkable duration for some tests, low reliability for others, or additional equipment needs. Even the most widely used test (MSCEIT) is not without difficulty [27]. Regarding the weaknesses that the EI evaluating instruments present, it is necessary to develop a new one that rectifies the problems found. Consequently, the Mobile Emotional Intelligence Test (MEIT) was developed, a web-based survey able to evaluate the many aspects covered by EI in a worthy and reliable way, bearing EI as ability. In 2013, an article was published that meant this present project's genesis [28]. Therein, the use of digital methodologies for EI evaluation was defended, inasmuch as it turns out to be a new model of obtaining data more rapidly, easily, and accessibly than the ones used so far.

Despite the fact that the development of this test is based on the theory of Salovey and Mayer [6]—according to which EI implies a set of four cognitive skills, named "branches", that serve to process emotionally relevant information—it was decided to exclude emotional facilitation. Due to this, the MEIT only counts emotional perception, emotional understanding, and emotional management. The reason for this decision is based on different studies that support the exclusion of the Facilitation branch through models that show better data [29–31]. Furthermore, the results of the eight MSCEIT subtests meta-analysis demonstrated quite high correlations among the factor of Facilitation and Perception, which caused the authors to recommend a three-factor model, obviating the facilitation branch [32]. MacCann, Joseph, Newman, and Roberts [33] defend that there exists a distinction between the second branch (emotional facilitation) and the other three, since perception and expression of emotions (first branch) as emotional understanding (third branch) and regulation (fourth branch) are related with the process of thinking about emotions, while the second branch (emotional facilitation) includes the use of emotions to ease thinking. For all these reasons, the facilitation branch was not evaluated in the MEIT.

Socioemotional problems tend to be too complex and ambiguous to justify only one type of answer as correct; thence, it was decided to classify the answer options in accordance to its adequacy instead of classifying them as "precise" or "inexact". Assuming that EI is a skill or set of skills rather than a personality characteristic [34], the more frequent answers are considered more appropriate than the infrequent. Moreover, given that experts in emotions are more prone to have a shared and accurate social representation of the correct answers [35], it was determined that the MEIT score be obtained by comparing the individual answers with the statistical distribution of the results taken by 32 experts in psychology and emotional field (17 men and 15 women). For that, all the possible answers in each item were considered by the proportion of experts that selected the answer. For instance, if 70% of experts chose answer "1" in an element, while 20% chose answer "2" and 10% answer "3", the individuals that chose answer "2" in this item would have a score of 0.20, while the participants that chose answer "1" would have a score of 0.70. In this study, the reliability among evaluators (coefficient of intraclass correlation) was 0.93.

In an attempt to offer a detailed analysis to the EI factorial structure, besides the factorial analysis, it was planned to provide empirical data about MEIT reliability and validity. In order to achieve this purpose, internal consistency was evaluated, as well as its convergent and divergent validity. Since some EI skill tests reveal discrepancies depending on age and genre, this type of group difference was also analyzed. In terms of EI convergent and divergent validity, it was intended to observe its relation to constructs that have been demonstrated to have a connection with EI:

**Fluid intelligence**. If EI represents a type of intelligence, the results of intelligence tests should correlate to the EI tests. Numerous previous studies have demonstrated that EI, through capability tests, is correlated in a significant way, but low with the fluid intelligence [36–38]. That correlation has to be moderated to exclude the possibility of constructs overlapping [39].

**EI feature (self-report).** Previous research confirms that the feature scales are incapable of predicting the results in the ability tests that they measure. The correlations between EI feature and EI ability are low [38,40], which shows that the first questionnaires evaluate aspects related to personality while the ability questionnaires evaluate aspects related to cognitive skills. Therefore, a significant, although weak, correlation is expected in EI self-reporting in the MEIT results.

**Personality**. The concerning results in the relationship between EI and personality are ambiguous, largely due to the different strategies of EI measurement used. Evaluating EI as a feature through self-reports leads to a considerable overlap between EI and the main five features of personality [41,42]. Nevertheless, when defining EI as ability no significant relationships should be expected with the main dimensions of personality. According to these bases, many empirical studies [38,43,44] have shown that EI evaluated as ability shares only a small fraction in the common variable in personality. For example, MSCEIT only correlated with openness (0.25) and kindness (0.28) [38].

**Life satisfaction**. The previous results confirm that there is a relation to EI, both for self-report scales and through ability tests. In the first case, the correlation between the EI score (measured through SREIT-Self-Report Emotional Intelligence Test) and psychological wellbeing (measured through Ryff's questionnaire) was 0.69, while the correlation between the EI score obtained through MSCEIT and Ryff's psychological wellbeing questionnaire was 0.25 [38]. For this, significant, although weak, correlations are expected between the life satisfaction questionnaire and the MEIT results.

**Gender.** Research conducted using self-reports shows different results about gender differences in EI [22,45,46], however when EI performance indicators are used, the result show that women obtain slightly higher scores than men [12,13]. It is expected that differences between genres are similar in the present study.

**Age**. Theoretically, if EI is an ability it should increase with age due to the acquisition of knowledge about emotions and the context in which they are developed [39,47]. However, other authors do not find significant association between age and EI ability [48,49]. According to these data, EI increases with time, but declines with age as any other cognitive ability [50]. It is probable that from childhood until adulthood people develop emotional abilities but that cognitive deterioration affects EI. Thus, small, although significant increases in each EI branch are expected until a certain age, as well as a small decrease in the older participants.

### 1.5. Aims of the Study and Research Hypothesis

The aim of this paper is to validate a new instrument to obtain an EI measure through tasks that consider emotional skills. For this, the MEIT inner structure and a wide net of convergent, discriminant, and psychosocial variables were analyzed. Based on previous research [26,33,34,51] it is expected that MEIT correlates significantly to fluid intelligence and wellbeing, however that results will differ from personality features and EI self-reporting, showing correlations close to 0. It is also predicted that higher scores will be found in women and adult participants.

## 2. Materials and Methods

### 2.1. Sample

We analyzed the data of 1549 people (841 women and 708 men) aged between 18 and 68, with an average age of 27.77 years (SD = 8.75). The sample was composed of 434 university students, 904 workers, 161 unemployed people, and 50 pensioners. Every subject has Spanish nationality. It is important to note that, in order to maximize validity, participants filled in questionnaires voluntarily and without economic reward; the lack of economic reward guaranteed that people who completed the tests were unbiased. The confidentiality of the data collected and the responsible use of the information taken from the investigation were reported. To examine the test–retest reliability, 140 people were selected randomly and surveyed at baseline and after two weeks. The design of this study was transversal and included EI aptitude measure, EI feature, fluid intelligence, life satisfaction, and personality.

### 2.2. Instruments

**Mobile Emotional Intelligence Test.** The MEIT evaluates perception, understanding, and emotional management, in that order, through 42 items in seven types of different tasks. To develop these tasks, the most empirically supported EI models were followed [52,53]. The first perceptive task (micro-gestures) consists of the identification of others' emotions. By means of six photographs of people expressing different moods, the user has to select, through four answer options on a 1-to-5 scale, the emotional degree that the person in each photograph is experiencing (Figure 1).

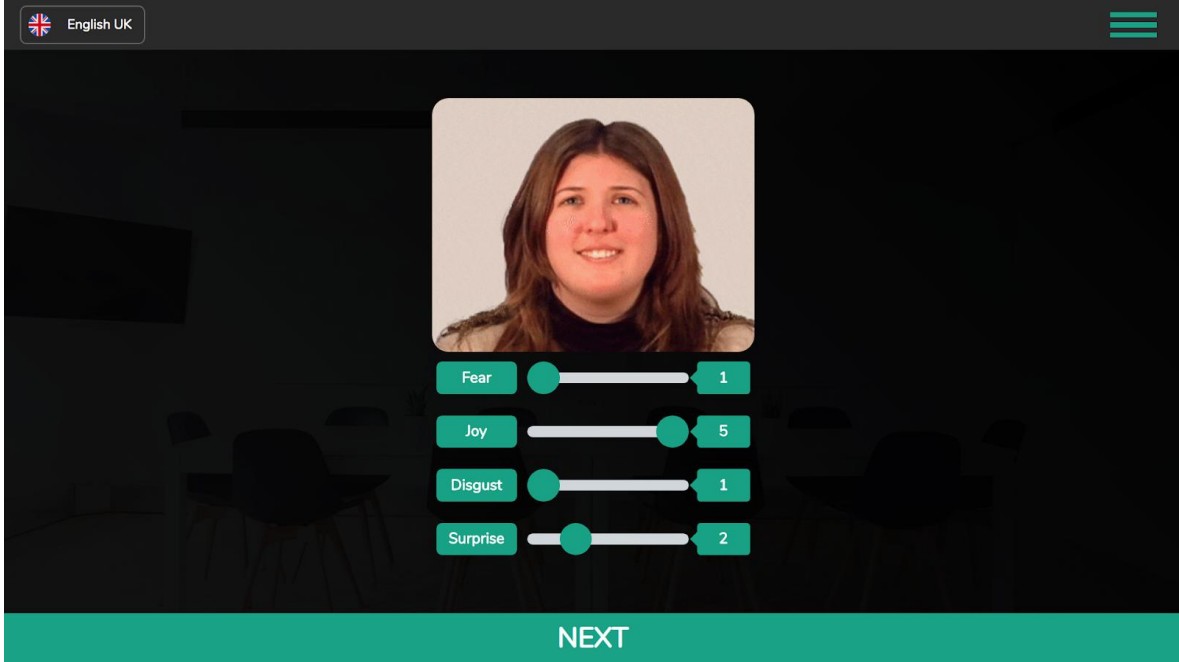

**Figure 1.** Micro-gesture task ("perception" branch).

In this sense, the approaches are diverse as well as the critics that have been carried out in the distinct ways of evaluating emotional perception [54,55]. In general, taking into consideration that emotions are the result of an interaction between the subject and the stimuli that surround him or her, it seems obvious that the evaluation of a subject's emotion should be performed resorting to instruments which allow emotion measuring at the moment when they are produced and not retrospectively. Thus, in this task the facial expressions were presented in a video form that allows observation of the change in facial gesture from the neutral status to the characterized status by a combination of determined emotions. This technical innovation raises ecological validity and the test's predictive value inasmuch as observing moving instead of static faces helps to perceive emotions in a similar way to that which is done in everyday life.

The second task in the perception branch is called "identification", where self-perception is evaluated. The user has to identify, on a scale of 1 to 10, how much intensity and wellness he or she feels when experimenting the shown emotion (Figure 2). In total, 10 distinct emotions are shown. For developing this task, Russell's classic "circumplex" model [56] was used as a base. There, emotions are considered to be a combination of two variables (energy and wellbeing), understanding energy as the neurological activation level and wellness as the (positive versus negative) valence.

The third perceptive task is named "faces" (three items with four images each one), and consists of selecting the most suitable photograph for the emotion shown in the superior part of the test (Figure 3).

The first understanding task (composition) is composed of three screens, and assesses the user's ability to understand how simple emotions evolve into more complex ones. For this, the subject is presented a name of a complex emotion together with a decanter, under which three test tubes represent simple emotions. The user then has to put the exact quantity of each emotion contained in the test tubes in the decanter to construct the indicated emotion (Figure 4). The development of this test was inspired by Plutchik's emotional taxonomic model [57].

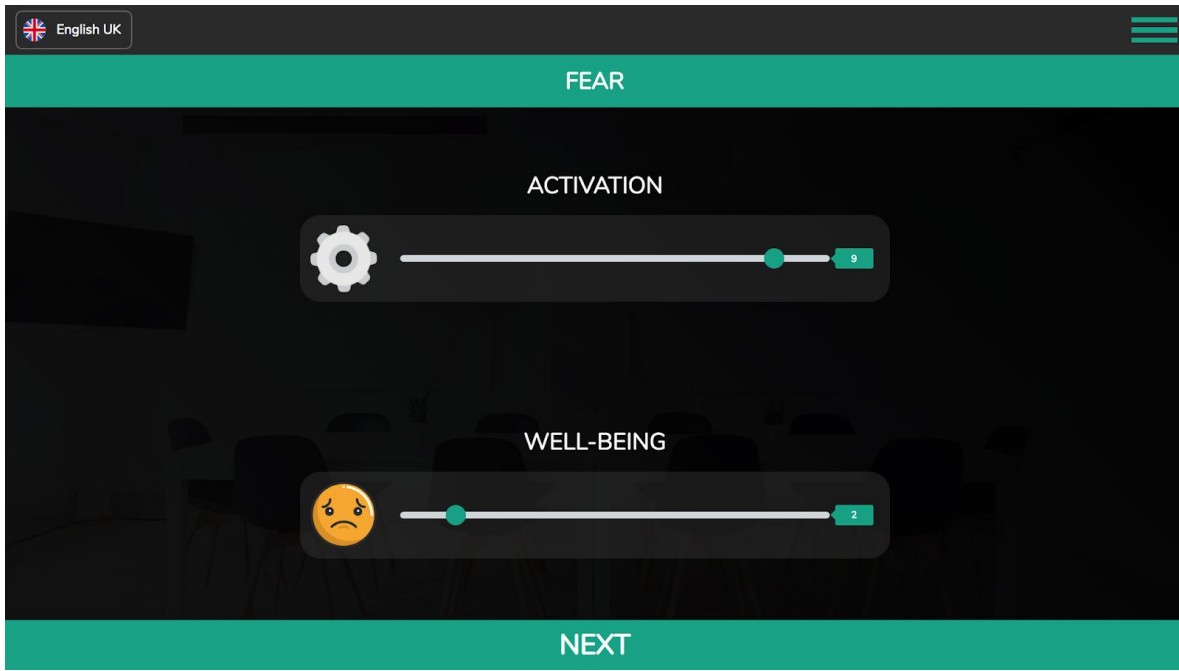

**Figure 2.** Identification task ("perception" branch).

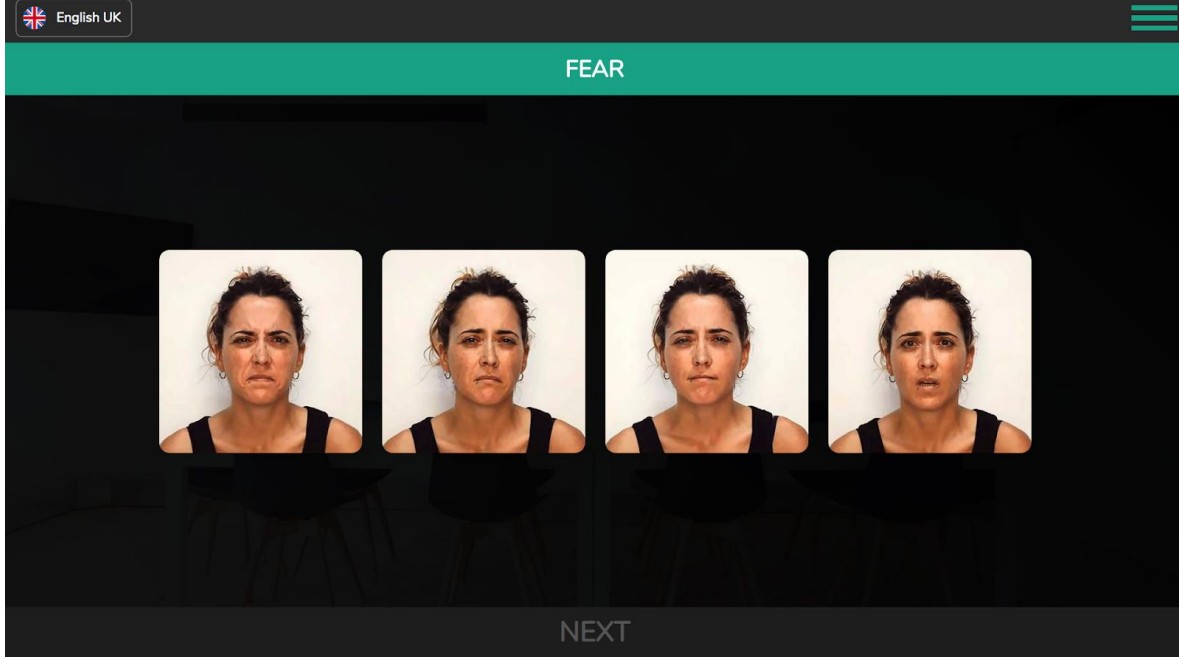

**Figure 3.** Faces task ("perception" branch).

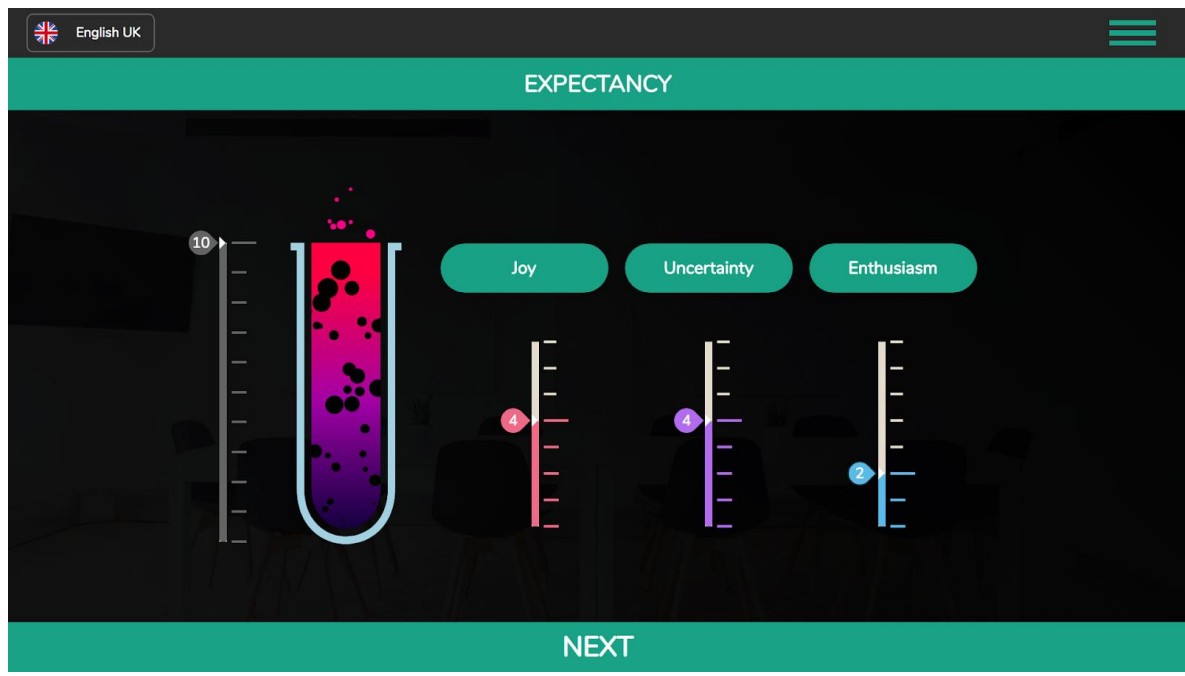

**Figure 4.** Composition task ("understanding" branch).

The second understanding task (deduction) evaluates people's ability to know how moods conduct to others depending on a described situation. In each of the three items that form this task, there is a story in which the main character lives a situation that causes a succession of emotions. The user is given two out of three emotions, and his or her aim is to guess the three among four answer options (Figure 5).

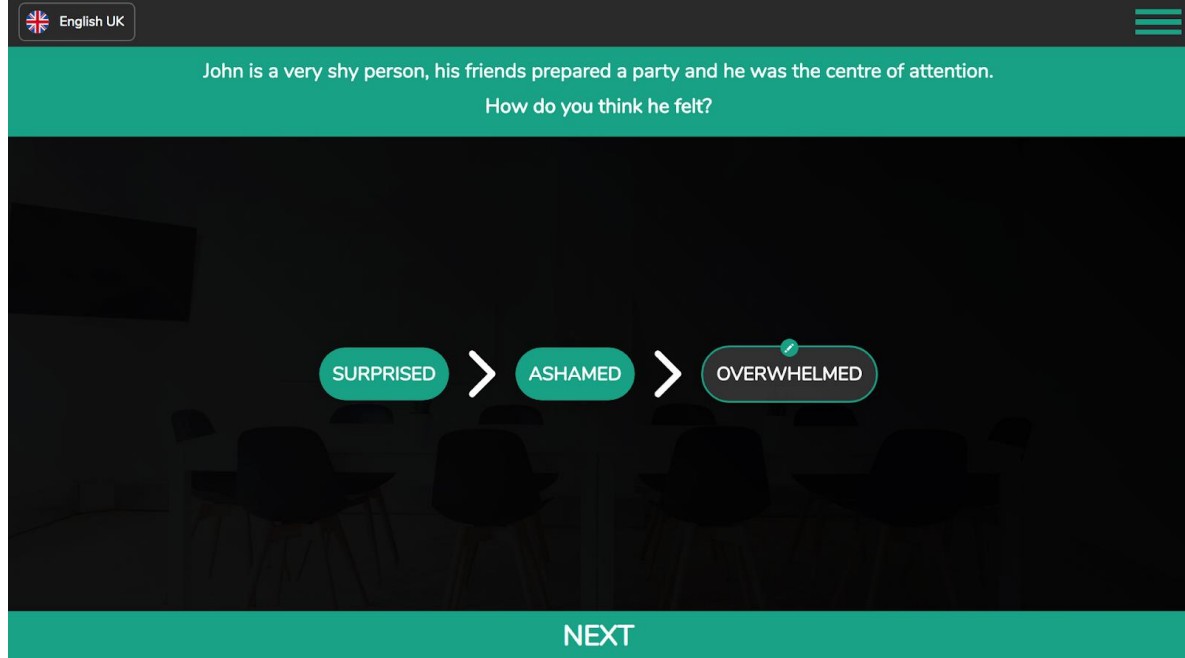

**Figure 5.** Deduction task ("understanding" branch).

The third understanding task (retrospective) is the same task as the previous one, but inverse; that is, the subject is given a series of emotions experienced by the main character and the subject has

to indicate, among three possible events, what he or she thinks may have happened to produce that emotion (Figure 6).

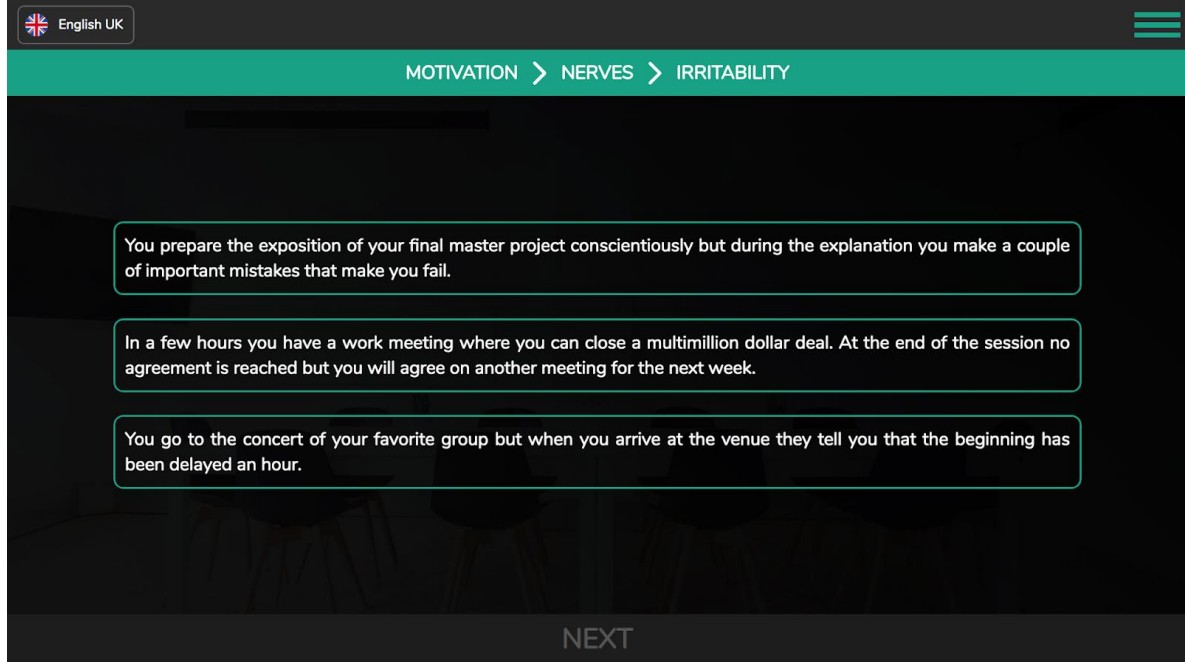

**Figure 6.** Retrospective tasks ("understanding" branch).

Lastly, the management branch is evaluated through a task (stories) composed of 14 items grouped in seven different situations, each one composed of two phases. The situations are classified according to the environment to which it makes reference (company, co-workers, and customers). The user reads a story that tells of a significant emotional event and has to indicate which is the best action to manage the indicated feelings (Figure 7).

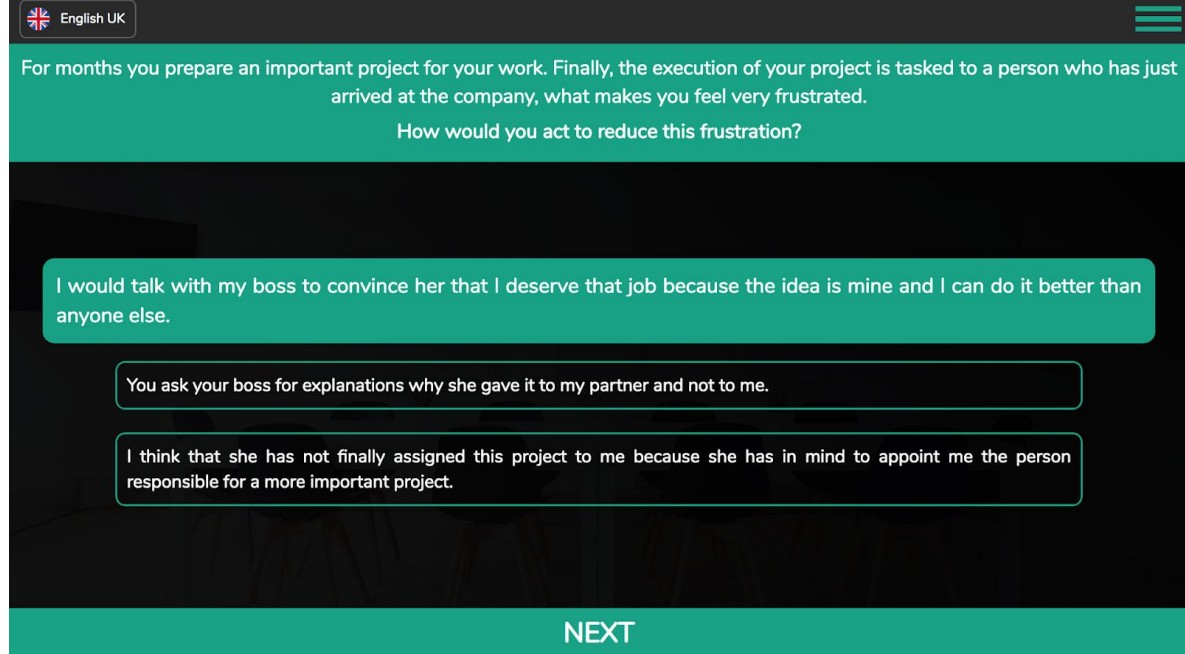

**Figure 7.** First step of stories task ("management" branch).

This branch is undoubtedly the most difficult to evaluate due to the challenge of measuring how people act in specific emotional situations in real life. After observing distinct possible solutions that allow achieving as much ecological validity as possible, it was decided that the best way of evaluating emotional management is making the user become completely involved in a story in which he or she is the main character and in which their decisions drive him or her to one situation or another. In this way, it is easier for the user to offer a real answer rather than one based on what he or she considers socially correct, since his or her answer has consequences that will be developed in the next question (Figure 8).

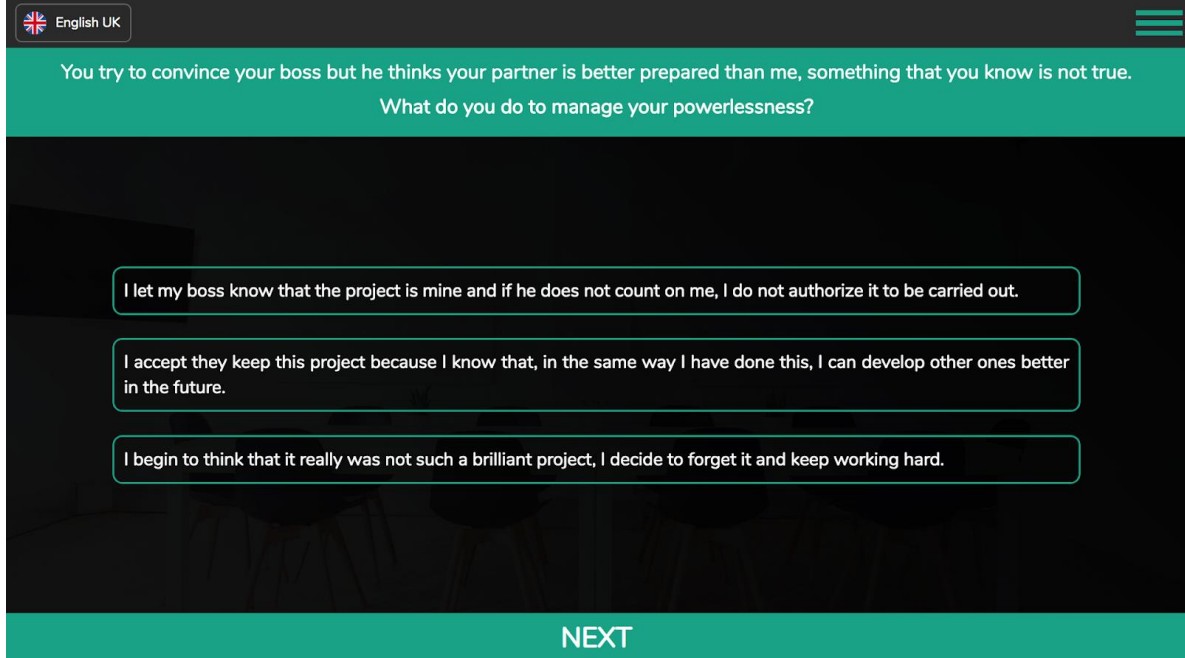

**Figure 8.** Second step of stories task ("management" branch).

**Trait Meta Mood Scale (TMMS-24)**. EI understood as feature has been evaluated with the TMMS questionnaire [16] in its reduced version translated into Spanish [58]. It is an EI self-report measure composed of 24 items with answers from 1 to 5. The tests assess three subscales (emotional attention, emotional clarity, and emotional repair), each one of which is composed of eight items, such as "I pay much attention to my feelings", "I am rarely confused about how I feel", or "I worry about being in too good a mood". Cronbach's alpha values for the three dimensions were 0.88, 0.86, and 0.90 respectively.

**RAVEN**. Raven's advanced progressive matrices tests [59] has been developed as a relatively efficient measure of general intelligence through geometric figure based on Spearman's g factor. In this study, a short form composed of 12 items was used, comparable to the larger form, developed by Arthur and Day [60]. The Cronbach's alpha value obtained in the present study was 0.78.

**Satisfaction With Life Scale (SWLS).** Developed by Diener, Emmons, Larsen, and Griffin [61], this scale is composed of five items, such as, "In most ways my life is close to my ideal", whose answer is given by a scale similar to Likert, from 1 (strongly disagree) to 7 (strongly agree). The SWLS Spanish version was used by the author in its web, and its Cronbach's alpha value was 0.81.

**Mini-markers**. Personality was measured through the test designed by Saucier [62]. This evaluates the five main personality traits using 40 descriptive adjectives, for example "talkative" or "shy", with which participants were self-marked in a nine-point scale from 1 (extremely inaccurate) to 9 (extremely accurate).

*2.3. Procedure*

This study conformed to the ethical guidelines mentioned in the Helsinki Policy Statements. All the process was carried out online in the year 2017. Firstly, participants were recruited from an online panel database already constructed by the research team. This database was formed of subjects who had participated in previous research not related to EI in different Spanish universities. A total of 8920 people were sent an email advertising the chance of participating in the study and providing details of its conditions. Secondly, those who decided to form part of the research were sent another email with the required information to complete the questionnaire fields and a link to enter the tests. It was recommended to work in silent environments, completing the tests without interruptions, and to answer individually. A total of 1607 subjects completed all the test fields. However, once the data-collecting process was finished, the final sample was composed of 1549 subjects, since all users who had response times of less than three standard deviations were deleted. For this, the time that subjects needed to read each item and the average response time to each item was obtained. The total average execution time was 1 h 26 min.

*2.4. Data Analysis*

The SPSS 24.0 statistical program (IBM) was used to calculate descriptive statistics, correlation analysis, internal consistency, and variance analysis. AMOS version 7.0 [63] was used for the confirmatory factor analysis (CFA). The goodness-of-fit of the models was evaluated using absolute and relative indices. Attending LISREL user guide version [64], the absolute goodness-of-fit indices calculated were: (1) The $\chi^2$ goodness-of-fit statistic; (2) The root-mean-square error of approximation (RMSEA); (3) The goodness of fit index (GFI); and (4) The adjusted goodness of fit index (AGFI).

The relative goodness-of-fit indices computed were: (1) the $\chi^2$ goodness-of-fit statistic; (2) The Root Mean Square Error of Approximation (RMSEA); and (3) The comparative fit index (CFI). The CFI is a population measure of model misspecification that is particularly recommended for model comparison purposes [65]. Non-significant values of $\chi^2$ indicate that the hypothesized model fits the data. Values of chi-square and RMSEA are to be smaller than 5 and 0.08, respectively. Values of CFI greater than 0.90 are considered as indicating a good fit [66].

## 3. Results

*3.1. Descriptive Statistics and Reliability*

Table 1 shows basic descriptive statistics for the MEIT, each of the three branches and each of the tasks that make up the test. Besides the average scores, the standard deviation, the kurtosis, and the skewness index are also depicted. The skewness and kurtosis results show distributions that can be treated as normally distributed. Additionally, the reliability values are also shown in Table 1. Following the procedures used in other cases to estimate the reliability of skill tests (MSCEIT), the correlation between the two halves was calculated. The MEIT full-test split-half reliability is 0.91, a result that is highly satisfactory. The three branch scores of Perceiving, Understanding, and Managing range between 0.77 and 0.92. The individual task reliabilities ranged from 0.71 to 0.91. Compared with the MSCEIT, reliabilities were very similar [21].

**Table 1.** Descriptive statistics and reliabilities for the Mobile Emotional Intelligence Test (MEIT).

| Total Score | Mean | SD | Kurtosis | Skewness | Reliability |
|---|---|---|---|---|---|
| MEIT | 26.67 | 4.27 | 0.24 | −0.11 | 0.91 |
| Perception | 16.1 | 2.39 | 0.21 | −0.02 | 0.92 |
| Micro-gestures | 5.89 | 0.75 | 0.08 | −0.06 | 0.91 |
| Identification | 5.69 | 0.55 | 0.09 | 0.02 | 0.87 |
| Faces | 5.17 | 0.45 | 0.11 | 0.03 | 0.84 |
| Understanding | 4.89 | 0.55 | 0.07 | −0.06 | 0.86 |
| Composition | 1.74 | 0.15 | 0.08 | −0.10 | 0.87 |
| Deduction | 1.49 | 0.25 | 0.06 | −0.06 | 0.85 |
| Retrospective | 1.55 | 0.48 | 0.07 | −0.08 | 0.79 |
| Management | 5.68 | 1.20 | 0.02 | 0.01 | 0.77 |
| Company | 1.75 | 0.48 | 0.09 | 0.05 | 0.71 |
| Coworkers | 1.98 | 0.57 | 0.06 | −0.02 | 0.73 |
| Customers | 1.95 | 0.58 | 0.03 | −0.01 | 0.77 |

## 3.2. Factor Structure and Intercorrelations

The one-factor model should load all eight MEIT tasks. The three-factor model loads the three designated tasks on each of the branches [21,52]. The confirmatory models shared in common that error variances were uncorrelated; latent variables were correlated, that is, oblique, and all other paths were set to zero. Both the general factor model and the three-branch model showed a good fit to the data. The adjustment parameters for the three-branch model were: $\chi^2$ = 1.406; CFI = 0.989; and RMSEA = 0.027. Figure 9 represents this model together with the standardized beta coefficients.

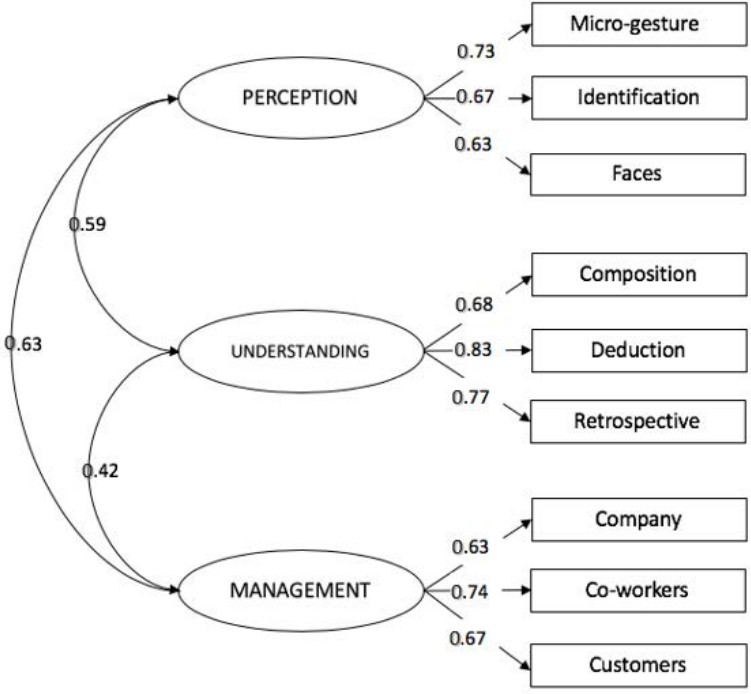

**Figure 9.** Results of the confirmatory factor analysis MEIT (i.e., perception, understanding, and management) (N = 1549).

As shown in Table 2, the correlational analyses revealed that branches of the MEIT are mutually intercorrelated and strongly related to the total score.

**Table 2.** Intercorrelations for the MEIT.

|  | MEIT Total | MEIT Perception | MEIT Understanding | MEIT Management |
|---|---|---|---|---|
| Perception | 0.78 ** | - | - | - |
| Understanding | 0.80 ** | 0.57 ** | - | - |
| Management | 0.83 ** | 0.60 ** | 0.65 ** | - |

** $p < 0.001$.

### 3.3. Validity (EI Trait, Personality, Intelligence, Satisfaction with Life)

In this section, the MEIT concurrent validity analyses are presented. To do this, correlation analyses were performed between the MEIT scores and the score in TMMS-24 (EI trait), mini-markers (personality), RAVEN matrices (Intelligence) and SWLS (satisfaction with life) (see Table 3).

**Table 3.** Correlations between MEIT and other tests.

|  | MEIT Total | MEIT Perception | MEIT Understanding | MEIT Management |
|---|---|---|---|---|
| EI (TRAIT) | 0.16 ** | 0.17 ** | 0.08 ** | 0.11 |
| Personality |  |  |  |  |
| Extraversion | 0.01 | 0.09 | 0.10 | −0.03 |
| Neuroticism | −0.01 | −0.07 | 0.04 | −0.08 |
| Openness | 0.03 | 0.08 | −0.03 | −0.05 |
| Conscientiousness | 0.09 | 0.01 | 0.08 | 0.12 |
| Agreeableness | 0.07 ** | 0.11 ** | 0.06 ** | 0.09 ** |
| Intelligence | 0.20 ** | 0.08 ** | 0.19 ** | 0.24 ** |
| Satisfaction with life | 0.18 ** | 0.09 ** | 0.17 ** | 0.21 ** |

** $p < 0.001$.

The total score of the MEIT correlated with self-reported EI at a significant but low level. Significant correlations were found between the MEIT subscales and personality traits. Weak, although significant, correlations with agreeableness, and the lack of any other relationships with the remaining "Big Five" dimensions of personality, should be interpreted as evidence that the MEIT does not cover preferences, habits, or inclinations. The total score in EI was positively and significantly associated with intelligence, while positive and significant correlations were found among all the subscales of the MEIT and intelligence. In this case, the moderate correlations between the MEIT and RAVEN suggest that EI is a set of mental abilities, and is related to intelligence. Finally, positive and significant correlations were found between the subscales of the MEIT and satisfaction with life.

### 3.4. Sex Differences and Correlations with Age

The main multivariate effect was significant for all branches (Wilk's lambda (3, 1587) = 24.96, $p < 0.0001$). In line with previous studies [12,13], women obtained higher scores than males in all the branches: perception of emotions, $F (1, 1780) = 46.44$, $p < 0.001$, d = 0.34; use of emotions, $F (1, 1780) = 43.61$, $p < 0.001$, d = 0.32; understanding of emotions, $F (1, 1680) = 36.35$, $p < 0.001$, d = 0.31; and the regulation of emotions, $F(1, 1780) = 81.40$, $p < 0.001$, d = 0.45. According to the criteria of Cohen [67], the effect size of these differences was small to moderate (<0.5). An analysis of variance was performed to analyze the gender differences in the total score. Women had a higher total than males: $F (1, 1780) = 95.67$, $p < 0.001$, d = 0.48. The size of the effect was moderate.

Regarding age, what was expected was confirmed. The correlations of Pearson showed low associations between the total MEIT and age (r = 0.09, $p$, 0.001.), as well as in the rest of the branches: perception (r = 0.06, $p$, 0.001), understanding (r = 0.11, $p$, 0.01), and management (r = 0.09, $p$, 0.001).

## 4. Discussion

This research was developed with the objective of discovering the psychometric properties of the MEIT, a new test of ability to evaluate EI. Generally speaking, the MEIT was found to be a reliable test of emotional intelligence. Thus, the hypotheses were confirmed.

Reliability measures in the MEIT subscales do not differ substantially from other tools of this type [68]. Many studies introducing new EI methods face the problem of weak convergent validity. The results seem to be in line with the empirical data published until now. The total score of the MEIT correlated with self-reported EI at a significant but low level oscillating around the threshold of a small effect size [67]. Compared with the convergent validity evidence, the discriminant validity evidence for the MEIT is promising. According to the ability-based approach, emotional intelligence is not a personality trait, and thus its measures should not share much variance with major personality dimensions. Many empirical studies [13,38,43,44] showed that EI shares only a small fraction of common variance with personality. In line with that, the MEIT proved its independence of personality. Weak, although significant, correlations with Agreeableness, and the lack of any other relationships with the remaining Big Five dimensions, should be interpreted as evidence that the MEIT does not cover preferences, habits, or inclinations. The moderate correlations between the MEIT and RAVEN suggest that EI is a set of mental abilities, related to intelligence, but independent of it. These findings are also consistent with EI literature [36–38].

Regarding gender differences, previous research shows that women tend to score higher than men on EI ability tests [12,13,69,70]. Also in the present study, women outperformed men in every single subscale of MEIT, and consequently in the total score. Such confirmation of the generally recognized phenomenon of women's advantage concerning emotional abilities is additional evidence supporting the MEIT validity.

According to Mayer and Salovey's theory, EI should increase with age due to the accumulation of knowledge about emotion and its social context (Mayer, Caruso, and Salovey, 1999; Burns, Bastian, and Nettelbeck, 2007). The results are in line with such prediction, and although the effect is very small and does not reach the small effect size threshold proposed by Cohen (1988), these results follow the line of previous investigations [30,39,47]. No evidence was found to support curvilinear changes in EI branches across lifespan.

## 5. Conclusions

In general, the instrument shows a clear internal structure, and its analysis reveals that the MEIT complies with the main criteria for assessing the validity of the EI test [34,50]. The empirical research confirms that the MEIT meets the psychometric rules related to reliability, validity (factorial structure), and discriminant validity. Nevertheless, the MEIT still presents some limitations. Future research should prove its reliability by examining, for example, whether the MEIT scores remain stable over time using a test–retest design and confirming that the test shows adequate reliability in other populations. It will also be necessary to explore the convergent validity of the MEIT with existing EI measures based on ability, such as the MSCEIT. A high correlation between the MSCEIT and the MEIT would definitely be adequate evidence for the convergent validity of the latter. Unfortunately, it was ruled out to use MSCEIT because it was not possible to integrate it digitally into the battery of tests and due to its long execution time.

Throughout this work, diverse EI evaluation instruments have been described. All of them have been validated and used in diverse research designs. However, EI evaluation methodologies are still undergoing a wide range of improvements [71]. For this reason, a new instrument has been presented, which combines characteristics of other tests that have already been published, adding some improvements that overcome the mentioned tests' weaknesses. It is important to consider the MEIT project innovation. Its technological character allows, firstly, facilitating its access, since users can access the tests anywhere, easing its administration. Moreover, thanks to this technological integration, the MEIT has advantages such as the possibility to register the user's answer time or to

incorporate videos in the development of the test. The MEIT also facilitates the date collection and its posterior analysis. It should not be denied that this type of platform helps to perceive the psychological evaluation as attractive, interesting, and motivating for both tested and tester. Thence, for example, in relation to perception, the MEIT combines fixed stimuli (photographs) that represent emotions, with dynamic stimuli (videos) in which the change from a neutral face to the emotion expression is appreciated. On the other hand, evaluating the understanding of emotions, the MEIT includes a combination of items in which basic emotions are used with other items to make complex emotions use. In terms of answer mode, items with enforced choice answer were combined with items in which the participant is asked to estimate the determined emotions' intensity. In this way, the possible bias that the use of the same analysis methodology has in the evaluation of human abilities has been minimized. Thus, the MEIT allows distinguishing not only those users that answer correctly among those that do not, but also those who answer rapidly among those who answer correctly. Besides, the MEIT is less vulnerable to the prejudices that affect the EI self-report measures, such as social desirability and answer style, as it is not based on self-perception and cannot be apparently falsified [13].

The MEIT was developed as a measure of an EI alternative ability. The main aim was to create an instrument to evaluate EI through skill tests that is easy to use and administrate and that does not require long time execution. The MEIT supposes a new approach about previous works. Primarily, it eases accessibility and the popularization of the evaluation of emotional perception without losing scientific rigor. Additionally, it increases the methodological rigor in this arduous task, including technological advances. All this may mean the first step towards a new dimension in the field of psychosocial evaluation: evaluation through mobile devices. The main goal has been accomplished. The MEIT is useful and valuable because it incorporates a distinctive set of characteristics which enrich the collection of available EI tests and will serve to advance the domain. In this sense, MEIT can be useful both for researchers and for educators who require a reliable and valid way to assess changes in EI, as well as to measure the impact of EI interventions.

**Author Contributions:** M.S.-G. and E.B. conceptualized the study and choose the theoretical framework. M.S.-G. and E.B. conceptualized the new scale and realized it. M.S.-G. collected the data and E.B. analyzed the data. Then the authors wrote the paper together and read and revised the manuscript several times.

**Funding:** This research was funded by Universitat Jaume I, grant number UJI-A2018-10.

**Acknowledgments:** We would like to thank Generalitat Valenciana and Fondo Social Europeo for providing co-funding to develop this research.

**Conflicts of Interest:** The authors declare that the research was conducted in the absence of any commercial or financial relationships that could be construed as a potential conflict of interest.

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
