# Peer review of "The Mobile Emotional Intelligence Test (MEIT): An Ability Test to Assess Emotional Intelligence at Work"

_sustainability, doi:10.3390/su11030827_

Round 1

Reviewer 1 Report

Thank you for giving me the opportunity of reading your paper. 

The topic of your paper is one a bit over-researched, but the novelty of the methods justify the publication. In general, your paper is well-written. 

Despite this fact, I would suggest some changes to improve it. 

Related to the theoretical background, some of your references are too old. I would suggest you update the literature to include the most recent papers on the topic. 

Related to the recruitment procedures, more information is needed. Which "online panel database" has been used? How has it been obtained? How are you sure that all the participants are Spanish people? Please, try to better explain this information for the readers. 

Related to the results, I would suggest you include the full matrix correlation with all the APA's characteristics. This is very relevant if your study will be included then in a meta-analysis. 

Your study has a lot of implications for individuals, but also for educators and policy-makers. Please, elaborate more on this point. 

Finally, some minor points: some references are only in Spanish. Please, check them. 

Author Response

Dear reviewer,

Based on the suggestions made by you, we have made a careful review of our paper proposal. In the new manuscript, we have used Word's change control to indicate the modifications or additions made to the original text.

We want to express our sincerest gratitude for your great work, your annotations have allowed us to improve the manuscript significantly.

Below we detail the changes made in the new version. We hope that the work done will achieve the final approval to be published. If not, we are at your disposal to resolve any issue or proceed with new revisions.

Sincerely,

The authors.

*************************************

Related to the theoretical background, some of your references are too old. I would suggest you update the literature to include the most recent papers on the topic.

Thanks for the contribution. We agree that some articles are old, we have incorporated several current articles that gather the latest advances in the subject. For example:

- Age and gender differences in ability emotional intelligence in adults: A cross-sectional study. Developmental Psychology 2016.

- Assessing the validity of emotional intelligence measures. Emotion Review 2018.

- The ability model of emotional intelligence: Principles and updates. Emotion Review 2016.

Related to the recruitment procedures, more information is needed. Which "online panel database" has been used? How has it been obtained? How are you sure that all the participants are Spanish people? Please, try to better explain this information for the readers.

Participants were recruited from a database obtained at the university (Universitat Jaume I) thanks to different research groups. This database was formed by subjects who had participated in previous research not related to emotional intelligence in different Spanish universities.

We are not sure that the participants have Spanish nationality, so we have decided to remove it from the text.

Related to the results, I would suggest you include the full matrix correlation with all the APA's characteristics. This is very relevant if your study will be included then in a meta-analysis.

It is true. We have added the complete matrix in APA format

Your study has a lot of implications for individuals, but also for educators and policy-makers. Please, elaborate more on this point.

Thank you so much. We have added a new paragraph in the discussion to contemplate the implications that MEIT can have on educators and new educational proposals

Finally, some minor points: some references are only in Spanish. Please, check them.

Thanks for the input. We have translated the three articles with Spanish title.

Reviewer 2 Report

First of all, I would like to thank you for having relied on me to carry out this review. An English style review is recommended.

The abstract must be prepared according to the following sections: Background, Methods, Results and Conclusion

Correct introduction, although personal writing should be avoided. Authors should use the indirect or passive form in this section and all the manuscript. For example line 86, 171, etc.

In the section on measures,  is necessary that authors include the Cronbach's alpha reliability, according to the sample collected (results obtained).

Has any ethical procedure been followed? Ethics committee? Authors must include it in the procedure section.

The results are adequately expressed and correct. It is recommended to make a possible analysis with the Factor program, to confirm the fit indices of the structural equation model.

The discussion should justify some of the assertions made by the authors. Are the authors' opinions critical? Nevertheless, what is shown in this section is correct, although a more current quotation is needed to discuss some relevant issues.

The bibliographical references do not fully conform to the Vancouver regulations, nor to the journal.

For example, the year's number should appear in bold

Author Response

Dear reviewer,

Based on the suggestions made by you, we have made a careful review of our paper proposal. In the new manuscript, we have used Word's change control to indicate the modifications or additions made to the original text.

We want to express our sincerest gratitude for your great work, your annotations have allowed us to improve the manuscript significantly.

Below we detail the changes made in the new version. We hope that the work done will achieve the final approval to be published. If not, we are at your disposal to resolve any issue or proceed with new revisions.

Sincerely,

The authors.

*************************************

First of all, I would like to thank you for having relied on me to carry out this review. An English style review is recommended.

Thank you for your time and dedication. We have made a revision to correct some errors in the text.

The abstract must be prepared according to the following sections: Background, Methods, Results and Conclusion

Thanks for the note. We have divided the sections better and we have added more information to the backgroud part.

Correct introduction, although personal writing should be avoided. Authors should use the indirect or passive form in this section and all the manuscript. For example line 86, 171, etc.

Thanks for the input. These expressions have been changed to impersonal forms.

In the section on measures,  is necessary that authors include the Cronbach's alpha reliability, according to the sample collected (results obtained).

Cronbach's alphas of all the instruments used have been added. We forgot to do it in the previous version.

Has any ethical procedure been followed? Ethics committee? Authors must include it in the procedure section.

We have add: This study was conformed to the ethical guidelines mentioned in the Helsinki Policy Statements. Additionally, the study was submitted to the evaluation and approval of the ethical committee of the Universitat Jaume I de Castelló (Spain).

The results are adequately expressed and correct. It is recommended to make a possible analysis with the Factor program, to confirm the fit indices of the structural equation model.

The fit indices of the structural equation model are in 3.2. Factor structure and intercorrelations.

The discussion should justify some of the assertions made by the authors. Are the authors' opinions critical? Nevertheless, what is shown in this section is correct, although a more current quotation is needed to discuss some relevant issues.

Thanks for the contribution. New articles have been incorporated into the discussion, furthermore, the implications that MEIT can have in educational environments and organizations have been deepened.

The bibliographical references do not fully conform to the Vancouver regulations, nor to the journal.

Thanks for the input. We have revised the rules of the magazine and we have corrected all the references.